# Traditional Chinese Medicine for Topical Treatment of Skeletal Muscle Injury

**DOI:** 10.3390/ph16081144

**Published:** 2023-08-12

**Authors:** Wing-Sum Siu, Hui Ma, Wen Cheng, Wai-Ting Shum, Ping-Chung Leung

**Affiliations:** 1Institute of Chinese Medicine, The Chinese University of Hong Kong, Shatin, New Territories, Hong Kong SAR, China; huima@cuhk.edu.hk (H.M.); wencheng@cuhk.edu.hk (W.C.); awtshum@cuhk.edu.hk (W.-T.S.); pingcleung@cuhk.edu.hk (P.-C.L.); 2State Key Laboratory of Research on Bioactivities and Clinical Applications of Medicinal Plants, The Chinese University of Hong Kong, Shatin, New Territories, Hong Kong SAR, China

**Keywords:** Chinese Medicine, topical treatment, muscle injury, muscle regeneration

## Abstract

Muscle injuries are common musculoskeletal problems, but the pharmaceutical agent for muscle repair and healing is insufficient. Traditional Chinese Medicine (TCM) frequently uses topical treatments to treat muscle injuries, although scientific evidence supporting their efficacy is scarce. In this study, an in vitro assay was used to test the cytotoxicity of a topical TCM formula containing Carthami Flos, Dipsaci Radix, and Rhei Rhizoma (CDR). Then, a muscle contusion rat model was developed to investigate the in vivo effect and basic mechanisms underlying CDR on muscle regeneration. The in vitro assay illustrated that CDR was non-cytotoxic to immortalized rat myoblast culture and increased cell viability. Histological results demonstrated that the CDR treatment facilitated muscle repair by increasing the number of new muscle fibers and promoting muscle integrity. The CDR treatment also upregulated the expression of Pax7, MyoD and myogenin, as evidenced by an immunohistochemical study. A gene expression analysis indicated that the CDR treatment accelerated the regeneration and remodeling phases during muscle repair. This study demonstrated that topical CDR treatment was effective at facilitating muscle injury repair.

## 1. Introduction

Muscle injuries, occurring in diverse professions and lifestyles such as athletics, manual labor and domestic activities, are a prevalent occurrence. A comprehensive report shed light on the substantial impact of musculoskeletal injuries, which account for 50.1% of the reported medical conditions among individuals aged 18 and over in the United States [1]. Notably, within this category, contusions alone contribute to 13% of the reported cases. Furthermore, it is noteworthy that over 90% of sports-related injuries manifest as either contusions or sprains [2].

Despite the high prevalence, muscle injuries are often regarded as minor problems in clinical practice. Patients typically receive conservative treatments [3] and are subsequently referred to physiotherapists for rehabilitation [4]. In terms of pharmaceutical treatments, analgesics for pain relief [5] and anti-inflammatory drugs to reduce swelling [6,7,8] are commonly prescribed. However, there is controversy surrounding the use of anti-inflammatory drugs, as inflammation may have positive effects on injury recovery and is necessary for skeletal muscle regeneration [9,10,11]. Prolonged use of anti-inflammatory drugs may extend inflammation inhibition and potentially impede the healing of muscle injuries. Therefore, the benefits of administering anti-inflammatory drugs for the treatment of muscle injury are under debate [12,13].

In Chinese communities, individuals often turn to Traditional Chinese Medicine (TCM) for remedies when they experience muscle injuries. TCM treatment approaches typically involve a combination of oral and topical medications, as well as techniques like massage and acupuncture. Among these, topical applications of TCM are widely preferred due to their non-invasive, safe and convenient nature. These applications come in various forms, including paste, patch, balm, liniment or oil [14,15]. Generally, adverse effects of these applications are rare, although instances of contact allergy and dermatitis have been occasionally reported [16,17]. However, many topical TCM products consist of multiple herbs, often ranging from 5 to over 20. This complex herbal mixing poses a challenge for scientists seeking to elucidate the underlying therapeutic mechanisms. Only a few of these products have been able to provide evidence-based scientific data to support their efficacy. Moreover, there is a lack of international publications reporting in vivo studies on the effects of topical TCM treatment specifically targeting skeletal muscle injuries. 

Traditional Chinese Medicine (TCM) encompasses a vast array of medicinal herbs, with over one hundred varieties specifically prescribed for injury treatment [18]. As scientific techniques and understanding have progressed, modern biological platforms have facilitated the evaluation of medicinal herbs to uncover their pharmacological activities. In light of this, herbs demonstrating angiogenic, anti-inflammatory, and cellular regeneration properties were screened in this study. Subsequently, we carefully selected the most popular and easily accessible herb from each category to compose a potent herbal formula. The three selected herbs were:(1)Carthami Flos (*Carthamus tinctorius* L.). A Chinese herb traditionally used for cardiovascular disease and bone injury in China with pharmacological effects on improving blood circulation. Its proangiogenic effect has been proven in both in vitro and in vivo studies [19].(2)Dipsaci Radix (*Dipsacus asperoides* C.Y. Cheng T.M. Ai). A common herb in traumatology that promotes blood circulation to remove a hematoma and alleviate pain. Recent scientific studies illustrated that Akebia Saponin D, the most abundant constituent of Dipsaci Radix, exhibits anti-inflammatory effects [20,21].(3)Rhei Rhizoma (*Rheum palmatum* Linn). A widely and traditionally used Chinese herb for wound healing. One of its derivatives, emodin, has been reported to promote the repair of rat excisional wounds through a complex mechanism involving the stimulation of tissue regeneration and regulation of the Smads-mediated transforming growth factor-β1 signaling pathway [22].

The amalgamation of the three carefully selected herbs resulted in an herbal paste aptly named CDR, derived from the abbreviation of their respective names. Clinical trials have demonstrated the efficacy of topically applying CDR for the treatment of conditions such as fifth metatarsal fractures and plantar fasciitis [23,24]. Furthermore, our research group has shown that when used in conjunction with oral pharmaceutical agents, CDR exhibits an additive effect, facilitating the healing process of fractures [25,26]. According to the principles of TCM, a similar or even identical topical TCM formula has traditionally been used to address minor injuries, including minor fractures, tendon and muscle injuries resulting from sprains, strains or contusions. Based on this knowledge, we hypothesized that CDR may also prove effective at treating muscle injuries.

The primary aim of this study was to investigate the effects of a topical three-herb formula on the healing process of muscle contusion injuries, both in vitro and in vivo. Furthermore, our objective was to unravel the underlying mechanisms involved. By doing so, we sought to provide scientific evidence to substantiate the utilization of TCM in topical treatments for skeletal muscle injuries.

## 2. Results

### 2.1. Effect of the Herbal Extract on Cell Viability of L6

The results of the MTT assay (Figure 1) showed that none of the herbal extracts of Carthami Flos, Dipsaci Radix, Rhei Rhizoma and CDR had a cytotoxic effect on L6 cells, even at a concentration of up to 1600 µg/mL. In fact, the extract of Rhei Rhizoma increased the L6 cell viability in a dose-dependent manner, with a significant increase of 11.60% (*p* = 0.0429), 21.17% (*p* = 0.0001) and 28.23% (*p* = 0.0001) for 400, 800 and 1600 µg/mL, respectively, starting from 400 µg/mL. Additionally, the CDR extract at a concentration of 1600 µg/mL also boosted the L6 cell viability by 13.87% (*p* = 0.0317).

### 2.2. Incapacitance Test

No significant change in the static weight ratio (SWR) of the rats was observed in any of the groups following contusion (Appendix A, Appendix A). Additionally, there was no significant difference in the SWR between the CDR and the Ctrl groups. These findings suggested that the degree of sensitivity in the incapacitance test was limited and incapable of distinguishing the level of pain experienced by the rats in the gastrocnemius between the two groups.

### 2.3. Morphological Change in Muscles after Contusion

The histological analysis revealed distorted architecture, ruptured and edematous muscle bundles, as well as dense diffuse infiltration of inflammatory cells in the contused gastrocnemius on Day 2 (Figure 2A,D,G). Seven days post-contusion, fibrotic tissue dominated the injured area and the presence of centronucleated myofibers indicated that muscle regeneration of the damaged muscle fibers had begun (Figure 2B,E,H). The CDR treatment, but not VTR, increased the number of new muscle fibers compared to the control group. By Day 14, there was evidence of clearing of necrotic tissue, and the area was being remodeled by normal tissue. The presence of centronucleated myofibers indicated that the regeneration of muscle fibers was still ongoing. Compared with the control group, muscle integrity was increased in both the CDR- and VTR-treated groups. Fascicles were formed primarily in the CDR and VTR groups but not in the control group (Figure 2C,F,I).

### 2.4. Immunostaining of Muscle

Immunofluorescent staining showed that the contusion significantly increased the number of Pax7^+^-stained nuclei per myofiber (Pax7 density) compared with the normal group (Figure 3). A significant increase was observed in the CDR group on both Day 2 (4.71-fold increase, *p* = 0.0014) and Day 7 (3.87-fold increase, *p* = 0.0109), as well as in the control group on Day 7 (3.33-fold increase, *p* = 0.0397). The Pax7 density in both groups then decreased by Day 14, with a significant decrease observed in the CDR group from Day 2 to Day 14 (*p* = 0.0391). Generally, the Pax7 density was higher in the CDR group, although not significantly, than in the control group at all time points, especially on Day 2 (1.60 folds, *p* = 0.1348). 

The immunohistochemical study demonstrated that none of the MyoD- and myogenin (MyoG)-positive nuclei were detected in the normal muscle, but they were significantly expressed after contusion injury (Figure 4 and Figure 5). The CDR treatment boosted both MyoD and MyoG expression at all time points, especially on Day 2, with the MyoD expression significantly higher than that of the Ctrl and VTR groups by 1.40-fold (*p* = 0.0378) and 1.44-fold (*p* = 0.043), respectively (Figure 4J). The expression of MyoD was highest on Day 2 and then decreased significantly by Day 7 and Day 14 post-injury in all groups (Figure 4J). The peak of the MyoG expression, however, was at Day 7 post-injury (Figure 5J), after which the MyoG level in the control and CDR groups decreased significantly by Day 14 (*p* = 0.043 and 0.0136, respectively).

### 2.5. Gene Expression

The expression of the genes related to muscle repair during the healing period was analyzed in the control group (Figure 6A). The results showed that *Pax7*, *Myod1* and *Myog* were significantly upregulated by 1.95-fold (*p* = 0.0039), 2.16-fold (*p* = 0.0005) and 2.42-fold (*p* = 0.0153), respectively, on Day 2 post-injury compared with the normal muscle, but it returned to normal levels thereafter. The peak expression of *Col1a1* and *Acta2* was observed on Day 7 post-contusion, where they were significantly higher than the normal muscle by 8.01-fold (*p* = 0.0006) and 2.23-fold (*p* = 0.0104), respectively. The expression of *Myh4* was highest on Day 14 post-contusion and was 1.95-fold higher (*p* = 0.047) than the normal muscle. 

In the inter-group comparison, the mRNA expression of *Pax7* and *Col1a1* in CDR was 1.65-fold and 2.43-fold higher, respectively, than in the control group on Day 2 post-contusion, and these differences were significant (*p* = 0.0139 and 0.0022, respectively) (Figure 6B). These two genes were also significantly higher in the CDR group than in the VTR group. The expression of *Pax7, Myod1, Myog* and *Myh4* in the CDR group was higher than in the control group on Day 7, although the difference was not significant (Figure 6C). Meanwhile, *Acta2* and *Vegfa* in the CDR group were higher than in the control group on Day 2 and Day 14 (Figure 6D), but without significant differences.

## 3. Discussion

Muscle regeneration encompasses a comprehensive process consisting of five distinct phases: degeneration/necrosis, inflammation, regeneration, remodeling and maturation/functional repair [27,28]. However, despite the significance of these phases, there is a notable scarcity of in-depth scientific studies published in international journals that explore the direct effects of the topical TCM on muscle injury repair, particularly during the latter three phases. A thorough search of the English language literature reporting in vivo studies utilizing various reputable search engines (such as PubMed, Science Direct, Scopus and Google Scholar) with specific keywords including “(Soft tissue OR muscle) AND (injury OR inflammation) AND (regeneration OR repair) AND (topical OR transdermal OR external) AND (Chinese Medicine)” yielded only five relevant publications [29,30,31,32,33]. Notably, these studies primarily focused on the anti-inflammatory, analgesic or anti-swelling effects of TCM on soft-tissue injuries. Consequently, researchers investigating the topical application of TCM for muscle injury treatment should prioritize these crucial phases as they represent the key steps in the overall muscle repair process. To achieve this purpose, it is imperative to conduct well-designed in vivo controlled studies. Such studies not only have the potential to validate the efficacy of TCM in facilitating muscle healing directly but also shed light on the morphological changes and the molecular mechanisms involved.

In the current study, an in vitro MTT assay was used to demonstrate the non-cytotoxic nature of the individual herbs comprising the CDR formula when tested on muscle cells. The results affirm the safety profile of CDR for use in muscle injury treatment. Furthermore, the application of CDR was found to significantly enhance muscle cell viability, indicating an increase in the number of live and healthy cells in response to extracellular stimuli, chemical agents, or therapeutic treatments. Notably, the observed effect on muscle cell viability was primarily attributed to the presence of Rhei Rhizoma, which exhibited a dose-dependent improvement in muscle cell viability. This finding aligns with the well-established role of Rhei Rhizoma in stimulating tissue regeneration, as described in the Introduction. The in vitro results were further supported by the histological analysis conducted during the in vivo experiment. The histological analysis revealed that CDR treatment led to an increased number of newly formed muscle fibers and improved muscle integrity after 7 and 14 days of contusion, respectively. Collectively, these results provide compelling evidence that topical treatment with CDR facilitates muscle regeneration.

By examining the genetic cascade involving paired box transcription factors (Pax3 and Pax7) and muscle regulatory factors (Myf5, MyoD, MyoG and MRF4) that regulate skeletal muscle regeneration, researchers can gain a comprehensive understanding of the intricate mechanisms underlying myogenic determination, cell differentiation and muscle regeneration. These factors provide valuable insights into each stage of satellite cell activation, the transient expansion of progenitor cells and the subsequent differentiation and formation of new muscle fibers [34]. 

The immunofluorescent staining conducted in the present study revealed a notable increase in the number of Pax7^+^ cells following the contusion, reaching its peak on Day 7. This finding aligns with previous reports by other researchers [35,36], further validating our observations. Additionally, our gene expression analysis using qPCR demonstrated a significant upregulation of *Pax7* following contusion, consistent with the findings reported by Tian et al. [36]. Pax7 serves as a key marker for satellite cells, playing a crucial role in maintaining their proliferation and preventing apoptotic cell death. While Pax7 can be detected during the quiescent state of satellite cells, its expression becomes more prominent once satellite cells are activated. 

MyoD, a transcription factor directly regulated by Pax7, plays a crucial role in satellite cell activation, proliferation and differentiation during muscle regeneration [37]. It exhibits an upregulation during the initial stages of muscle regeneration [38]. Consistent with the findings of Tian et al. [36], our IHC and qPCR results demonstrated a significant upregulation of MyoD in parallel with Pax7 following skeletal muscle injury. These results collectively indicate the activation of muscle regeneration following contusion injury. Additionally, MyoG, a muscle-specific transcription factor, is involved in the coordination of skeletal muscle development and repair processes. In normal muscle, MyoG-positive nuclei are not detectable, but they appear three days after injury and increase further by Day 7 as myogenic progenitors (myoblasts) differentiate into myocytes [38,39]. Our IHC analysis of MyoG corroborated these observations. 

Inconsistencies were observed in the results between the IHC and qPCR analyses of gene expression, particularly when comparing different groups. Notably, on Day 2 post-injury, the CDR group exhibited a higher number of MyoD^+^ and MyoG^+^ nuclei per myofiber compared to the Ctrl and VTR groups (Figure 4J and Figure 5J, respectively). However, CDR was less effective than VTR at inducing *Myod* expression. Although *Myog* expression was reduced in both the CDR and VTR groups, the difference was not statistically significant (Figure 6B). These discrepancies could potentially be attributed to differences in the specific regions investigated during the two assessments. The IHC results were derived from the images captured specifically within the central damaged zone, whereas the qPCR analysis involved harvesting the entire contused region of the gastrocnemius muscle, encompassing the damaged zone, border zone and the non-damaged tissue. In a separate study, the findings from in situ hybridization demonstrated that while both *Myod* and *Myog* mRNA were expressed post-injury, the density of MyoD and MyoG staining was highest at the periphery of the damaged site rather than at the central damaged site [40].

Significantly, the findings of the current study revealed that CDR treatment resulted in a notable increase in the number of Pax 7^+^, MyoD^+^ and MyoG^+^ nuclei, as well as their corresponding gene expression level when compared to the Ctrl group. In contrast, the effect of the VTR treatment was observed to be similar to that of the Ctrl group. These observations strongly support the notion that topical CDR treatment effectively expedites the muscle regeneration process following contusion injury. 

Type I and III collagen play a pivotal role as the predominant collagens in the intramuscular connective tissue, forming an essential network within the extracellular matrix (ECM) [41]. ECM serves as the scaffold for the establishment of novel myofibers and neuromuscular junctions. The active deposition of ECM components closely accompanies the connective tissue remodeling process during muscle healing. Notably, Type I collagen deposition occurs later than Type III during the formation of the connective tissue scar, with its expression peaking after 7 days post-injury [42]. Coinciding with this timeline, our qPCR analysis demonstrated a significant upregulation of *Col1a1* gene expression on Day 7 following contusion. Similarly, the qPCR analysis of *Acta2*, a gene encoding the vascular tissue in skeletal muscle, revealed a significant upregulation after contusion, reaching its peak on Day 7. This finding suggests that the revascularization rate in our contused rat model was highest on Day 7. Remarkably, the CDR treatment resulted in increased expression of *Col1a1*, *Acta2* and *Vegfa* on Day 2. This observation indicates that CDR treatment accelerated fibrogenesis and angiogenesis and shifted the remodeling phase of muscle regeneration forward during muscle repair effectively.

Additionally, the detection of markers such as adult myosin heavy chain (MyHC) isoforms serves as a valuable indicator for the mature phenotype of newly regenerated muscle fibers [27]. In our study, the gene expression level of *Myh4* exhibited a significant peak on Day 14 following contusion, signifying the advanced stage of muscle regeneration. Notably, topical CDR treatment demonstrated a trend toward upregulating *Myh4* expression compared with the Ctrl group, although statistical significance was not achieved. These results strongly suggest that topical CDR treatment also enhances the maturation phase of the healing muscle, further promoting the development of functionally mature muscle fibers.

The primary objective of this study was to investigate the impact of CDR on the mechanism underlying muscle regeneration, without assessing the functional recovery of the contused muscle. It is important to acknowledge that there are certain limitations to this study. Firstly, the specific in vivo effects of each individual herbal component on muscle regeneration were not examined, thus preventing us from determining their individual contributions. Additionally, the potential synergistic effects resulting from the combination of these herbs could not be conclusively demonstrated. These aspects, including the individual in vivo effects, synergistic effects and in-depth mechanistic pathways underlying the observed results, will be addressed in future studies.

## 4. Materials and Methods

### 4.1. Herbal Materials and Authentication

All the herbs were purchased from Guangzhou Zhixin Limited (Guangzhou, China) and authenticated using thin-layer chromatography, as reported previously [43]. The herbarium voucher specimens of the tested herbs were deposited in the museum at the Institute of Chinese Medicine, the Chinese University of Hong Kong, with voucher numbers as follows: Carthami Flos (2013-3415); Dipsaci Radix (2013-3417); Rhei Rhizoma (2013-3416). The herbal extracts were prepared according to a previous report [43]. Briefly, each herb was extracted individually with refluxing in distilled water. The remaining solid herbal residue was further extracted with refluxing in 95% ethanol. The aqueous and ethanol filtrates were mixed and concentrated into paste form. The CDR paste was formed after mixing the three individual herbal pastes in a dry weight ratio of 1:1:1. 

The chemical composition of the CDR paste was determined using high-performance liquid chromatography-electrospray ionization-mass spectrometry (HPLC-ESI-MS). The system used in this analysis included an Agilent 1290 Infinity LC System (Agilent Technologies, Santa Clara, CA, USA) and an Agilent 6410 Triple Quad LC/MS equipped with an ESI source. A Waters HSS T3 3.5 µm (3.0 mm × 150 mm) HPLC column maintained at 40 °C was utilized. Six chemical markers (Appendix A in the Appendix A) and CDR samples were separated using a gradient mobile phase consisting of water (A) and acetonitrile (B). The gradient condition was 0 to 3 min, 10–27% B; 3 to 5 min, 27–33% B; 5 to 12 min, 33–33% B; 12 to 13 min, 33–80% B; 13 to 16 min, 80–90% B; and 16 to 20 min, 90–90% B. The flow rate was set at 0.5 mL/min, and the injection volume was 20 µL. The MS analysis was monitored in negative ion mode and multiple reaction monitoring mode using target ions at m/z 611.2→325.0 for hydroxysafflor yellow A; *m/z* 285.0→117.0 for kaempferol; m/z 927.5→603.3 for asperosaponin VI; *m/z* 455.3→407.4 for oleanolic acid; *m/z* 269.0→241.0 for emodin; and *m/z* 283.0→239.0 for rhein. The abundance of the chemical markers was determined quantitatively (Appendix A, Appendix A). All the chemicals were purchased from Sigma-Aldrich (St. Louis, MI, USA), and both methanol and acetonitrile were HPLC-grade (≥99.9%). The chemical markers were purchased from Tauto Biotech Co., Ltd. (Shanghai, China).

For the in vitro study, the pastes were dissolved in a relative culture medium and filtered through a 0.22 µm filter. For the in vivo topical treatment, the CDR paste was supplemented with 2.0% (*w*/*w*) borneol (Alfa Aesar, Shanghai, China) to increase the transdermal efficiency.

### 4.2. In Vitro Study 

The rat myoblast L6 cell line was purchased from the American Type Culture Collection (ATCC, Manassas, VA, USA). A total of 5 × 10^4^ cells/well in DMEM growth medium supplemented with 10% FBS and 1% penicillin–streptomycin (Life Technologies, Carlsbad, CA, USA) were seeded in each well of a 96-well culture plate, with 200 µL L6 per well. The cells were incubated in a 37 °C incubator with 5% CO_2_ and 95% humidified air overnight. The medium was then replaced with 200 µL DMEM containing herbal extract of either Carthami Flos, Dipsaci Radix, Rhei Rhizoma or CDR at concentrations ranging from 0 (control) to 1600 µg/mL. The cells were further incubated for 24 h.

Cell viability of the L6 was assessed using the 3-(4,5-dimethylthiazol-2-yl)-2,5-diphenyltetrazoliumbromide (MTT) assay. A total of 20 µL MTT solution (5 mg/mL in PBS) was added to each well containing 200 µL medium and incubated for 4 h at 37 °C. The resultant formazan product was dissolved in DMSO (100 µL/well) and measured at 540 nm using a microplate spectrophotometer. The mean of six experiments was calculated for analysis.

### 4.3. Animal Model and In Vivo Study

The contusion model, which involves inducing injuries on the gastrocnemius muscle of rats, is a highly recognized and frequently used model for studying the intricate process of skeletal muscle injury repair [44]. Animal ethics approval was obtained from the Animal and Experimental Ethics Committee of the Chinese University of Hong Kong (CUHK) for the in vivo study (13/058/MIS). A total of 54 male Sprague–Dawley rats with a mean body weight of 353.3 ± 17.7 g, supplied by the Laboratory Animal Service Centre (LASEC), CUHK, were used. The rats were housed in standard cages at a constant temperature of 22 °C with a 12 h light–dark cycle and given ad libitum access to food and water. The experimental procedures were commenced after 7 days of acclimatization. 

The rats were anesthetized using ketamine and xylazine cocktail (im), and buprenorphine (sc) was administered for analgesic purposes. The rats were placed supine with their right legs fixed in position. A contusion injury was created on the right leg of the rats from the medial–lateral approach (posteromedial position). To administer the contusion impact, a cylindrical stainless steel blunt weight weighing 1 kg, with an 8 mm contact diameter, was dropped from a height of 15 cm onto the gastrocnemius muscle, ensuring no impact on the tibia. Subsequently, one gram of the CDR herbal paste (CDR) was applied topically to the posterior lower hindlimb and covered with an adhesive gauze pad. In the positive control group, the CDR herbal paste was substituted with one gram of Voltaren^TM^ (VTR) gel. It contains diclofenac sodium as the active ingredient and is commonly prescribed by physicians for the treatment of muscle injuries. The application of the herbal paste and the gel was renewed daily, five days a week, over a period of three weeks. As an experimental control, rats that did not receive either the CDR paste or the VTR gel were utilized (referred to as the Ctrl group). The animals in each group were euthanized under anesthesia at 2, 7 and 14 days post-injury, with 6 animals in each time point in each group. The contralateral limb of each animal without contusion and treatment served as the normal control (Norm).

### 4.4. Incapacitance Test

An incapacitance test was utilized to determine the level of pain or the analgesic effect of the treatments on the animals. Only the rats terminated at Day 14 of the Ctrl and CDR groups were evaluated to validate the reliability of this test in the current study. Rats were constrained in an acrylic plastic holder to stand on the two sensors of the incapacitance meter (Panlab Harvard Apparatus, Holliston, MA, USA) calmly and steadily. The weight of each hindlimb was measured separately by the two sensors as the static weight. The static weight ratio (SWR), which is the ratio of the injured leg to the Norm, was calculated to eliminate the interference of the body weight changes throughout the study [43]. 

### 4.5. Histological Assessments 

The gastrocnemius muscle of the rats was harvested from the injured site at selected times post-injury (2, 7, and 14 days post-injury). The muscle was embedded in optimal cutting temperature compound (O.C.T.) (Tissue-Tek) and frozen by submerging it in isopentane pre-cooled with liquid nitrogen. Cryosections of 8 µm were obtained and stored at −20 °C until use. The cryosections were stained with Hematoxylin and Eosin (H&E) using routine methods to evaluate the morphological changes in the muscle during healing. Immunohistochemical and immunofluorescent staining were also performed for quantitative analysis.

#### 4.5.1. Immunofluorescence

Immunofluorescence (IF) staining was carried out for Pax7 and laminin. First, the sections were fixed with pre-cooled acetone for 10 min followed by blocking with 10% normal goat serum (G9023, Sigma-Aldrich) for 1 h at room temperature. They were then incubated with rabbit anti-laminin (ab11575, Abcam, Cambridge, UK, 1:500) overnight at 4 °C, followed by incubation with the secondary antibody goat anti-rabbit IgG conjugated with Alexa Fluor^®^ 594 (ab150080, Abcam, 1:200) and then blocked with goat F(ab) anti-rabbit IgG (ab6824, Abcam, 1:1000) at room temperature for 1 h separately. Subsequently, the sections were co-stained with rabbit anti-Pax7 (PA5-68506, Invitrogen, Carlsbad, CA, USA, 1:200) overnight at 4 °C, followed by incubation with the secondary antibody goat anti-rabbit IgG conjugated with Alexa Fluor^®^ 488 (ab150077, Abcam, 1:200) at room temperature for 1 h. Finally, the sections were mounted with an aqueous mounting medium containing DAPI (sc-24941, Santa Cruz Biotechnology, Dallas, TX, USA). Three animals from each group were included in this analysis.

#### 4.5.2. Immunohistochemistry

Immunohistochemistry (IHC) was performed using the DAB (3,3′-diaminobenzidine) method (K500711-2, Dako REAL EnVision Detection System, Agilent Technologies, Santa Clara, CA, USA). The procedures suggested by the manufacturer were followed. Briefly, the sections were fixed followed by blocking as mentioned for the IF above. Then, they were incubated overnight at 4 °C with the primary antibodies, including mouse anti-MyoD (MA1-41017, Invitrogen, 1:500) and mouse anti-MyoG (MA5-11486, Invitrogen, 1:500). After washing, the sections were incubated at room temperature with the secondary antibody and the DAB substrate of the DAB kit for 30 and 1 min, respectively. At least three animals from each group were included in this analysis.

A negative control for the immunostainings (IF and IHC) was included using staining without the primary antibodies.

#### 4.5.3. Image Analysis 

In the H&E and IHC analyses, the sections were visualized using an upright microscope (Eclipse Ci-L, Nikon, Tokyo, Japan) at a magnification of 400×. The images were captured using a digital camera (DS-Ri2, Nikon, Tokyo, Japan). For the IHC, the number of nuclei stained positively with MyoD or MyoG, as well as the number of myofibers per region of interest, were measured using image software (ImageJ, US National Institutes of Health, Bethesda, MD, USA). The number of positively stained nuclei was then normalized with the number of myofibers. 

In IF, fluorescence images were captured using an inverted fluorescent microscope (IX71, Olympus Corporation, Tokyo, Japan). Five ROIs from each section at the magnification of 400× were captured using a digital camera (DS-Fi3, Nikon, Tokyo, Japan). The number of nuclei stained positively with Pax7 and the number of myofibers per ROI were measured using ImageJ. The number of Pax7-stained nuclei normalized with the number of myofibers was calculated.

### 4.6. Gene Expression on Muscle Regeneration

The mRNA was extracted from the contused gastrocnemius muscle using an RNeasy Mini kit (Qiagen, Hilden, Germany) and reverse-transcribed into cDNA using Omniscript RT kit (Qiagen) with oligo-dT primers (Life Technologies, Carlsbad, CA, USA). For the quantitative real-time PCR (qPCR), an ABsolute QPCR Mix SYBR Green kit (Thermo Fisher, Waltham, MA, USA) was used with a Light Cycler (Bio-Rad Laboratories Inc., Hercules, CA, USA). The fold changes in mRNA expression were calculated using the 2^−ΔΔCt^ method [45]. First, the change in the mRNA expression of *Pax7*, *Myod1*, *Myog*, *Myh4*, *Col1a1*, *Acta2* and *Vegfa* after the contusion (within Ctrl) was analyzed. Then, their expressions were compared among different groups at different time points. The gene glyceraldehyde 3-phosphate dehydrogenase (*Gapdh*) was used as the reference gene. The primer sequences of the genes are listed in Table 1.

### 4.7. Statistical Analysis

Data are presented as mean ± standard error of the mean (SEM), unless otherwise specified. A two-way ANOVA followed by Tukey’s multiple comparisons test was used to compare data among groups and time points from the incapacitance test, immunofluorescent and immunohistochemical staining. For the in vitro assay and gene expression analysis, a one-way ANOVA was used, followed by Dunnett’s or Tukey’s multiple comparisons test, using GraphPad Prism 7. A *p*-value less than 0.05 was considered statistically significant among the groups.

## 5. Conclusions

Topical TCM treatment is widely utilized for muscle injuries within the Chinese community. However, its recognition in Western countries is hindered by the lack of support from the international scientific literature. Consequently, there is a pressing need for additional experimental studies rooted in scientific evidence to substantiate the efficacy of TCM in facilitating muscle repair following injury. The current study provides valuable insights by demonstrating that topical CDR treatment effectively enhances the regeneration, remodeling and maturation phases of muscle regeneration subsequent to injury.

## Figures and Tables

**Figure 1 pharmaceuticals-16-01144-f001:**
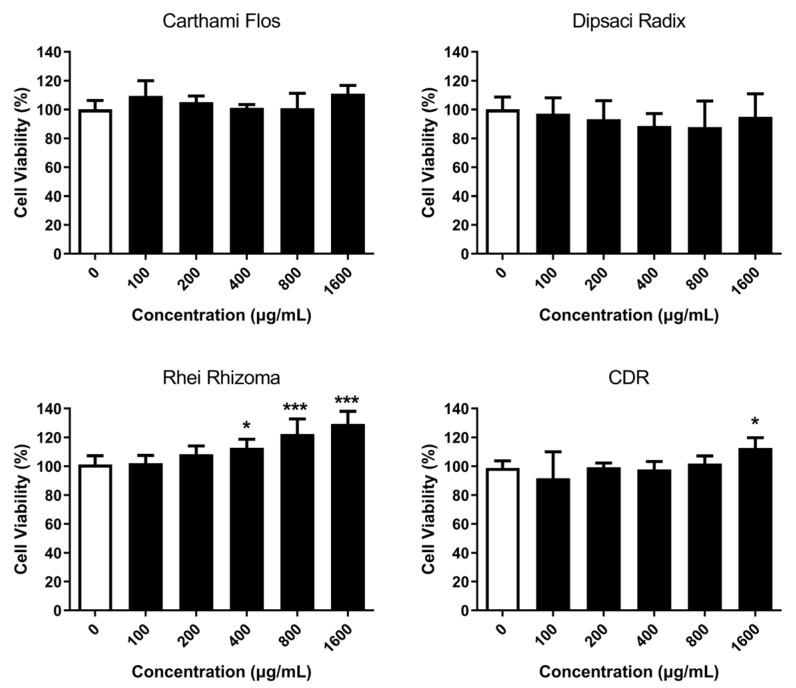
The viability of L6 cells in response to the herbal extracts and their combination (CDR) was evaluated. The white bar represents the control group, where cells were treated without any herbal extract, while the black bars represent cells treated with herbal extract at different concentrations (x-axis). Data are presented as the mean of percentage, with error bars indicating the standard deviation. Statistical significance is indicated using * for *p* < 0.05 and *** for *p* < 0.001 compared with the control group. The sample size for each group was n = 6.

**Figure 2 pharmaceuticals-16-01144-f002:**
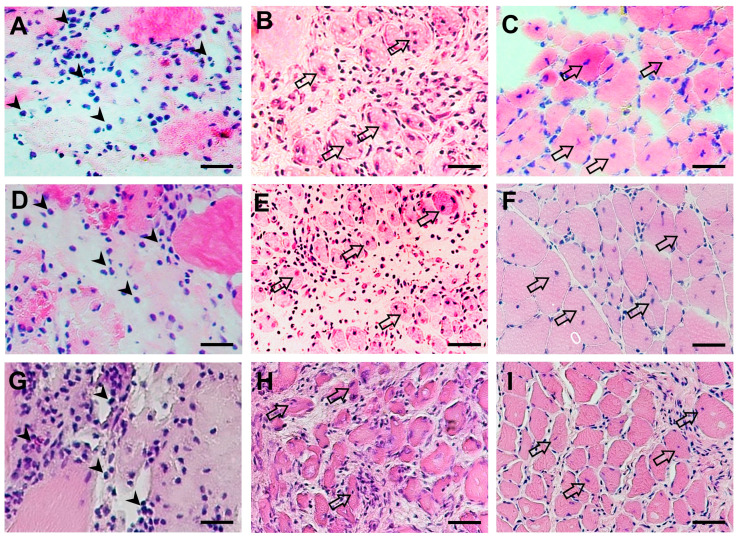
The morphological changes in the muscle after contusion in the different treatment groups were evaluated. Muscle sections were stained with H&E, and images were captured on Day 2 (**A**,**D**,**G**), Day 7 (**B**,**E**,**H**) and Day 14 (**C**,**F**,**I**). The upper panel shows the control group, the middle panel shows the CDR group and the lower panel shows the VTR group. Inflammatory cell infiltration is indicated using arrowheads, while centronucleated myofibers are indicated using hollow arrows. The images were captured at a magnification of 400×, and the scale bar represents 50 µm.

**Figure 3 pharmaceuticals-16-01144-f003:**
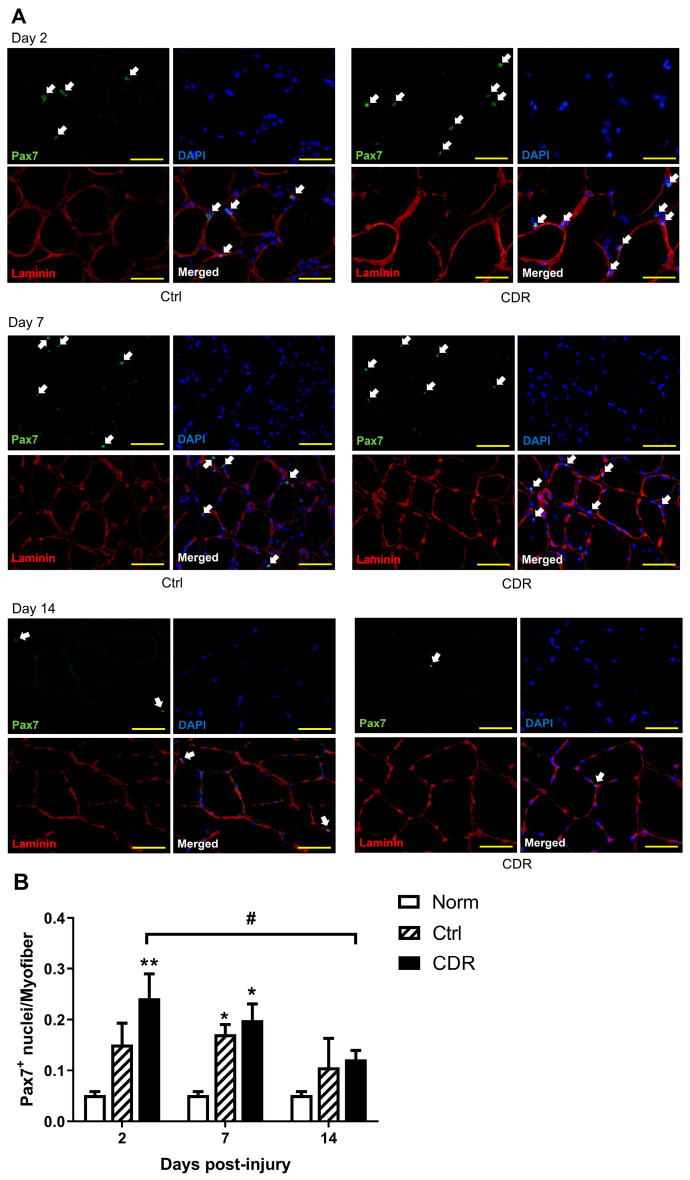
The change in Pax7 density in the muscle after contusion was evaluated. (**A**) Immunofluorescent staining of Pax7. Arrows indicate the Pax7^+^ nuclei. The images were captured at a magnification of 400×, and the scale bar represents 50 µm. (**B**) The quantitative results are presented for different groups at different time points post-injury. The error bars indicate the standard error of the mean, and statistical significance is indicated using * for *p* < 0.05 and ** for *p* < 0.01 compared with the normal group at each time point. The horizontal bracket indicates statistical significance with # for *p* < 0.05 when comparing between two different time points within the same group.

**Figure 4 pharmaceuticals-16-01144-f004:**
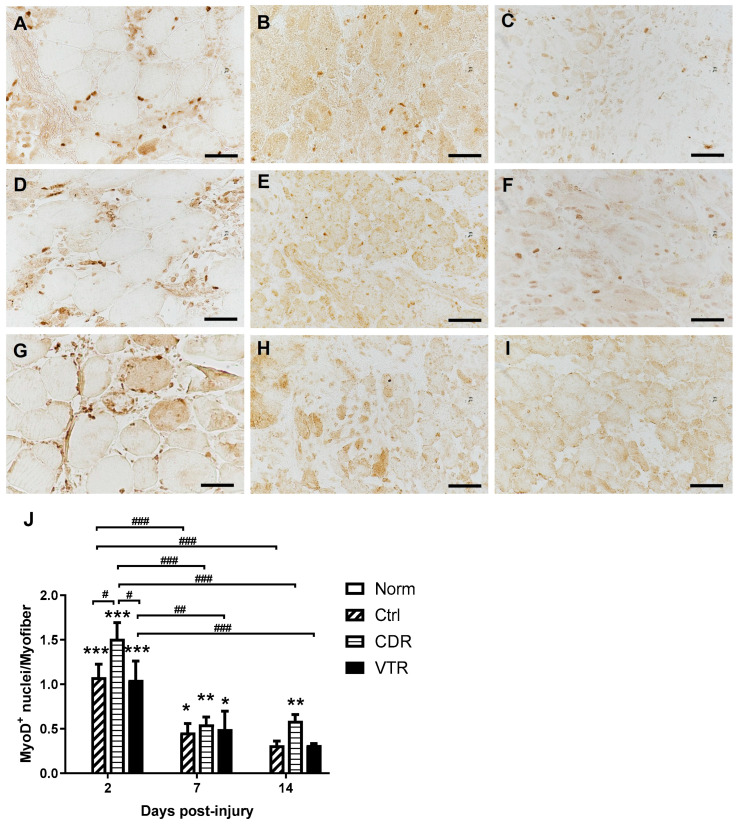
The change in MyoD density in the muscle after contusion was evaluated. A: immunohistochemical staining of the muscle (upper panel: Ctrl; middle panel: CDR; lower panel: VTR) on Day 2 (**A**,**D**,**G**), Day 7(**B**,**E**,**H**) and Day 14 (**C**,**F**,**I**). Nuclei were stained in dark brown color. The images were captured at a magnification of 400×, and the scale bar represents 50 µm. (**J**) The quantitative results are presented for different groups at different endpoints post-injury. The error bars indicate the standard error of the mean, and statistical significance is identified using * for *p* < 0.05, ** for *p* < 0.01, and *** for *p* < 0.001 compared with the normal group. The horizontal bracket indicates statistical significance with # for *p* < 0.05, ## for *p* < 0.01, and ### for *p* < 0.001 when comparing two different groups. It should be noted that the MyoD density in the normal group was zero, so the bars are not included in the figure.

**Figure 5 pharmaceuticals-16-01144-f005:**
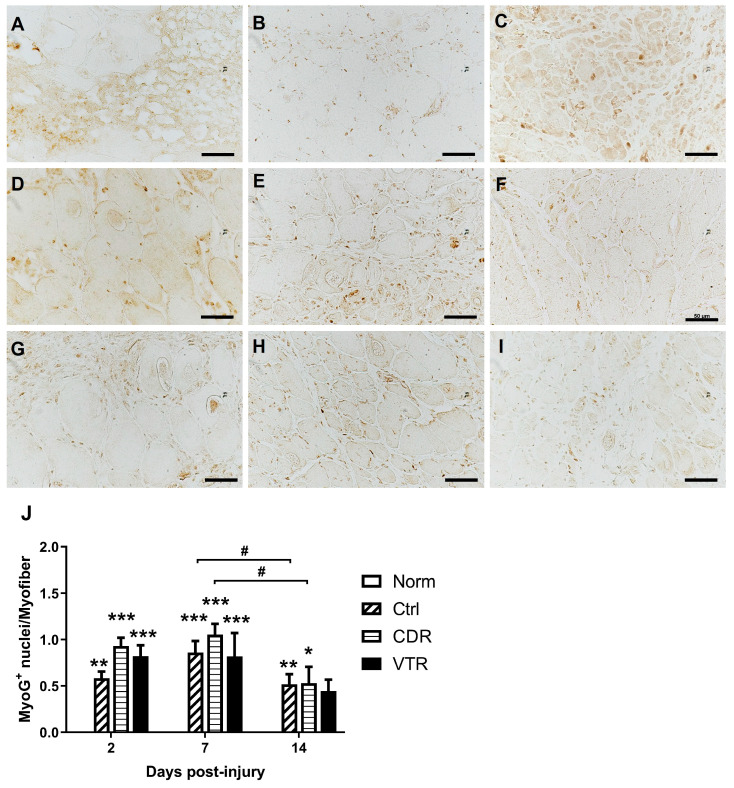
The change in MyoG density in the muscle after contusion was evaluated. A: Immunohistochemical staining of the muscle (upper panel: Ctrl; middle panel: CDR; lower panel: VTR) on Day 2 (**A**,**D**,**G**), Day 7(**B**,**E**,**H**) and Day 14 (**C**,**F**,**I**). Nuclei were stained in dark brown color. The images were captured at a magnification of 400×, and the scale bar represents 50 µm. (**J**) The quantitative results are presented for different groups at different endpoints post-injury. The error bars indicate the standard error of the mean, and statistical significance is indicated using * for *p* < 0.05, ** for *p* < 0.01, and *** for *p* < 0.001 compared with the normal group. The horizontal bracket indicates statistical significance with # for *p* < 0.05 when comparing two different groups. It should be noted that the MyoG density in the normal group was zero, so the bars were not included in the figure.

**Figure 6 pharmaceuticals-16-01144-f006:**
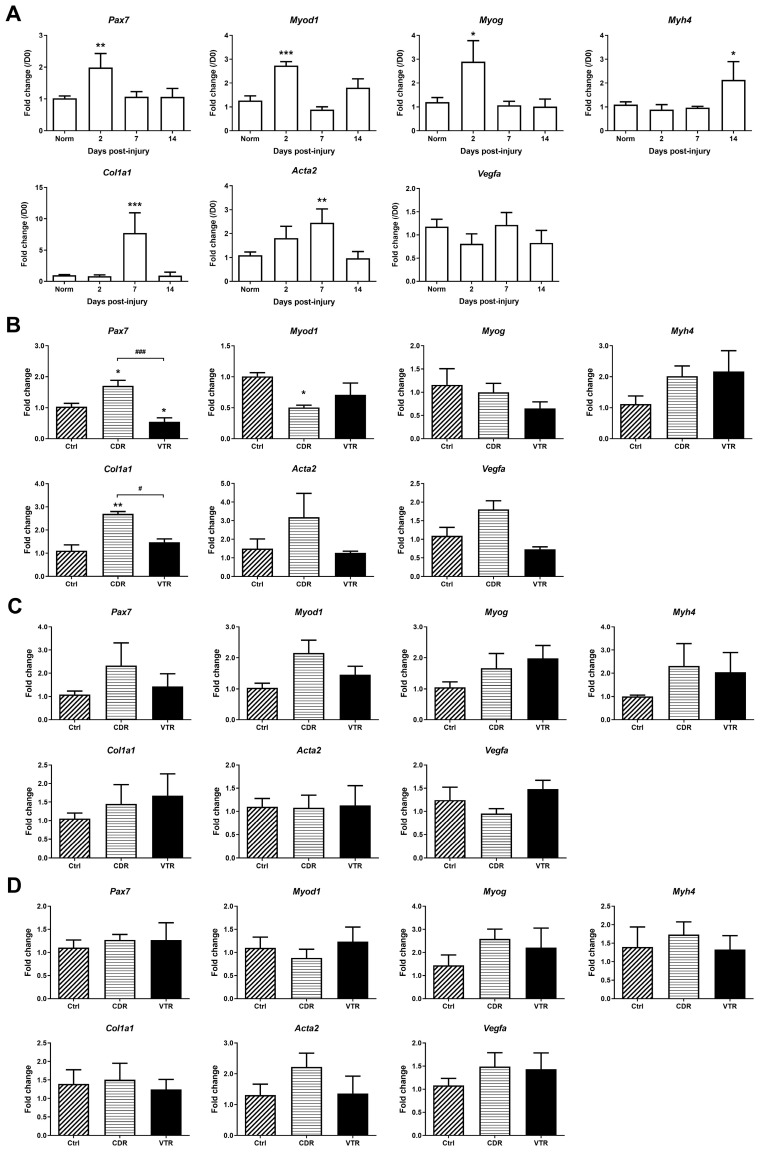
The mRNA expression of genes in muscle regeneration was analyzed. Panel (**A**) shows the change in gene expression before and after contusion. Panels (**B**,**C**,**D**) show the gene expression among different groups on Day 2, Day 7 and Day 14, respectively. The error bars represent the standard error of the mean, and statistical significance is indicated using * for *p* < 0.05, ** for *p* < 0.01, and *** for *p* < 0.001 compared with the normal group in Panel (**A**) or the control group in Panels (**B**–**D**). The horizontal bracket indicates statistical significance with # for *p* < 0.05 and ### for *p* < 0.001 when comparing two different groups.

**Table 1 pharmaceuticals-16-01144-t001:** Primer sequences of the genes.

Gene	Sequence (5′ to 3′)	NCBI Accession No.
*Acta2*	Forward: AACACGGCATCATCACCAACT Reverse: TTTCTCCCGGTTGGCCTTA	NM_031004.2
*Col1a1*	Forward: CCCAGCGGTGGTTATGACTT Reverse: GGGTTTGGGCTGATGTACCA	NM_053304.1
*Gapdh*	Forward: CTCAGTTGCTGAGGAGTCCC Reverse: ATTCGAGAGAAGGGAGGGCT	NM_017008.4
*Myod1*	Forward: GGAGACATCCTCAAGCGATGC Reverse: AGCACCTGGTAAATCGGATTG	NM_176079.1
*Myog*	Forward: GACCCTACAGGTGCCCACAA Reverse: ACATATCCTCCACCGTGATGCT	NM_017115.2
*Myh4*	Forward: CACACCAAAGTCATAAGCGAA Reverse: CCTTGATATACAGGACAGTGA	NM_019325.1
*Pax7*	Forward: GATTAGCCGAGTGCTCAGAATCAAG Reverse: GTCGGGTTCTGATTCCACGTC	NM_001191984.1
*Vegfa*	Forward: TACCTCCACCATGCCAAGTG Reverse: TCTGCTCCCCTTCTGTCGTG	NM_031836.3

## Data Availability

The data are contained within this article and the Appendix A.

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
