# Peer review of "Traditional Chinese Medicine for Topical Treatment of Skeletal Muscle Injury"

_pharmaceuticals, 2023, doi:10.3390/ph16081144_

Round 1

Reviewer 1 Report

The study by Sui et al. reports on the efficacy of traditional Chinese medicine (TCM) by topical treatment of skeletal muscle injuries. Sui et al. utilized in vitro and in vivo models, performed a comprehensive set of experiments, and generated inclusive data that support their final conclusions. Thus, I acknowledge the relevance of the outcomes and the scientific soundness of the report. Unfortunately, the manuscript is not well written and would improve from English language editing. Materials and Methods lack some relevant information. There are also inconsistencies and misunderstandings in the diagrams and figure legends.

Here are few examples:

1) Title can be improved: Traditional Chinese Medicine for Topical Treatment of Skeletal Muscle Injury

2) Abstract: The abbreviation “CDR” is not explained.

3) Introduction: The abbreviation “CDR” and “VTR” should be explicated.  

4) Results: The figure legends are in adequate. Black and white bars are not described in the legend to figure 1. Do the data represent a single experiment of repeated experiments? How many times? Mean values? Does100% refer to control? Figure 4: Why are P values indicated by asterisks and hashtags? What means “specified with the n-zig-zag line”?

The authors are kindly advised to revise the manuscript text and figures carefully.

Extensive editing of English language required.

Author Response

Comments and Suggestions for Authors

The study by Sui et al. reports on the efficacy of traditional Chinese medicine (TCM) by topical treatment of skeletal muscle injuries. Sui et al. utilized in vitro and in vivo models, performed a comprehensive set of experiments, and generated inclusive data that support their final conclusions. Thus, I acknowledge the relevance of the outcomes and the scientific soundness of the report. Unfortunately, the manuscript is not well written and would improve from English language editing. Materials and Methods lack some relevant information. There are also inconsistencies and misunderstandings in the diagrams and figure legends.

Here are few examples:

1) Title can be improved: Traditional Chinese Medicine for Topical Treatment of Skeletal Muscle Injury

Response: Thanks for the comment and suggestion. The title has been revised accordingly.

2) Abstract: The abbreviation “CDR” is not explained.

Response: The abbreviation “CDR” has been explained in the Abstract.

3) Introduction: The abbreviation “CDR” and “VTR” should be explicated.  

Response: Thanks for the comment. The abbreviation “CDR” has been explicated in Introduction. The abbreviation of “VTR” has been explained in Materials and Methods where it firstly appeared after considering the fluency of the manuscript.

4) Results: The figure legends are in adequate. Black and white bars are not described in the legend to figure 1. Do the data represent a single experiment of repeated experiments? How many times? Mean values? Does100% refer to control? Figure 4: Why are P values indicated by asterisks and hashtags? What means “specified with the n-zig-zag line”?

Responses: In Figure 1, the White bar was the control (i.e. the cells treated without herbal extract (0ug/ml)) and the Blacks bars represented the cells treated with the herbal extracts in different concentrations as indicated on the X-axis. I labeled the Control bar in white color (100% cell viability) to make it easier to be identified. I have clarified this in the figure legend. The result in Figure 1 was obtained from 6 repeated in vitro experiments. This information was mentioned in the last sentence in Section 2.2. (Originally Section 4.2). The bar was the mean and this information was also mentioned in Section 2.7. (Originally Section 4.7). However, “Data was presented as mean of percentage” and “n=6” were added in the revised figure legend.

In Figure 4 (Figure 3 in after revision), the p values indicated by asterisks and hashtags because they were obtained from two statistical comparisons. The p values indicated by asterisks were obtained when the Ctrl and CDR were compared with Norm at each time point. The p value indicated by hashtag was obtained when the data within the same group from different time points was compared. The “n-zig-zag line” is the name of the horizontal line indicating the two groups being compared in the Prism statistic software. To make it easier to be understood by readers, the figure legend has been revised.

The authors are kindly advised to revise the manuscript text and figures carefully.

Response: Thank you for your comments and we have revised the manuscript text and figures extensively.

Comments on the Quality of English Language

Extensive editing of English language required.

Response: We have edited the English language of the manuscript extensively.

Reviewer 2 Report

This work tests the effects of topical treatment by herbal paste, called CDR, on skeletal muscle regeneration using a rat contusion injury model. By histological examination of muscle sections, immunostaining of myogenic factors, and RT-qPCR analyses of gene expression, the authors conclude that CDR treatment facilitates muscle repair. Although this study is potentially interesting, it is preliminary, and the results are not consistent in some experiments.

1. The authors do not explain why they use CDR instead of individual herbal extracts in muscle regeneration, because figure 1 shows that Rhei Rhizoma extracts are more potent than CDR to induce cell viability. It is thus unclear for the readers what are the benefits of using CDR. The authors may need to test the extracts in muscle regeneration individually.

2. There is no information regarding CDR in the introduction section.

3. The formation of new myofibers was only examined in histological sections (figure 3). It is not obvious to see centralized nuclei in these images. Embryonic myosin light chain antibody should be used to identify regenerating myofibers. Combined with DAPI staining, the authors should also measure fusion index to quantify the efficiency of CDR on muscle regeneration.

4. There are also other problems with figure 3. The regenerating zones are not outlined, and it is not clear whether identical areas were used to compare the numbers of myofibers with centralized nuclei. In addition, images in figure 3b, e, and h were not properly stained, and it is not possible to see myofibers.

5. The rational to compare CDR and VTR in muscle regeneration is not clear.

6. In figures 4 and 5, it is not clear how many regenerating myofibers were examined in each condition.

7. The authors should show immunostaining of MyoD and MyoG in regenerating myofibers.

8. There are some inconsistent results in RT-qPCR analyses of gene expression. Figure 5A shows that CDR induces more MyoD-positive nuclei than VTR at 2 days post-injury. However, figure 6B shows that CDR is less efficient to induce MyoD expression than VTR. Similarly, figure 5B shows more MyoG-positive nuclei at 2 days after CRD and VTR treatment, but figure 6B shows reduced expression of MyoG in both conditions.

9. Figure 2 should be moved to supplementary materials, while figure 4S should be shown in the main text, but the quality and magnification of the images should be improved.

The rational for some experiments should be better explained, and some descriptions in the method section need to be revised.

Author Response

Comments and Suggestions for Authors

This work tests the effects of topical treatment by herbal paste, called CDR, on skeletal muscle regeneration using a rat contusion injury model. By histological examination of muscle sections, immunostaining of myogenic factors, and RT-qPCR analyses of gene expression, the authors conclude that CDR treatment facilitates muscle repair. Although this study is potentially interesting, it is preliminary, and the results are not consistent in some experiments.

  1. The authors do not explain why they use CDR instead of individual herbal extracts in muscle regeneration, because figure 1 shows that Rhei Rhizoma extracts are more potent than CDR to induce cell viability. It is thus unclear for the readers what are the benefits of using CDR. The authors may need to test the extracts in muscle regeneration individually.

Responses:

In TCM, herbs are seldomly used individually. It is because (1) in TCM theory, a disease will make disorders in different “systems” in our bodies simultaneously. For instance, it may cause inflammation, slow down the blood circulation and impair the immune system at the same time. TCM practitioners will issue a TCM formula containing different ingredients which aim at treating each disorder specifically; (2) there will be synergistical effect between different herbs so that the efficacy of a formula will be higher than a single herb; (3) some herbs are used to “neutralize” or attenuate the adverse effect of the other major herbs in the formula. A brief explanation of using CDR formula in this study has been added in the Introduction.

The main purpose of the in vitro study was to show that each herbal item and their combination are not cytotoxic to muscle cells. They are safe to be used to treat muscle disorders. This has been stated in the 2nd paragraph of the Discussion. In addition, the in vitro result also demonstrated that Rhei Rhizoma extracts are more potent than CDR to induce cell viability. It echoes its main function in the formula: to stimulate tissue regeneration. This explanation has been added in the 2nd paragraph of the Discussion.

We accepted that, from scientific points of view, we need to test the extracts in muscle regeneration individually. This was one of the limitations in the current study and it had been stated in the second last paragraph of the Discussion. However, the main objective of this study is to verify the efficacy of herbal formula (CDR) on muscle regeneration. The focus of the experimental design was therefore put on the CDR. When the in-depth mechanisms of the CDR are going to be studied in the future, we will test the extracts individually.

  1. There is no information regarding CDR in the introduction section.

Responses: Thanks for the comment. Information of CDR has been added in the Introduction.

  1. The formation of new myofibers was only examined in histological sections (figure 3). It is not obvious to see centralized nuclei in these images. Embryonic myosin light chain antibody should be used to identify regenerating myofibers. Combined with DAPI staining, the authors should also measure fusion index to quantify the efficiency of CDR on muscle regeneration.

Responses:

Thanks for the comment. Figure 3 (Figure 2 after revision) has been replaced by images in a higher magnification to show the centralized nuclei more obviously.  

We also thanks for your suggestion of using embryonic myosin light chain antibody to identify regenerating myofibers. However, our objective of using adult myosin heavy chain (Myh4) was to distinguish, if any, the phenotype of regenerated myofibers among different groups. We considered it as a maker at the late stage of the remodelling phase of muscle regeneration [Ciciliot and Schiaffino, 2010]. Another literature has also reported that the mature phenotype of newly regenerated muscle fibers can be highlighted by the presence of markers including adult myosin heavy chain (MyHC) isoforms [Ref 30: Forcina et al, 2020].

For the suggestion of measuring “fusion index”, we agreed that it is a good parameter. However, there may be some drawbacks. One of them is that it is difficult or impossible to be used to determine in situations of nuclear clustering or overlapping, two common features of cultured myotubes [Agley et al, 2012].

Taken together, we would like to keep our results but revised Figure 3 in this manuscript. The suggestions of the reviewer will be deeply considered in our future studies.

References:

Ciciliot S, Schiaffino S. Regeneration of mammalian skeletal muscle. Basic mechanisms and clinical implications. Curr Pharm Des. 2010;16(8):906-14. doi: 10.2174/138161210790883453.

Forcina L, Cosentino M, Musarò A. Mechanisms Regulating Muscle Regeneration: Insights into the Interrelated and Time-Dependent Phases of Tissue Healing. Cells. 2020 May 22;9(5):1297. doi: 10.3390/cells9051297. (Citation 30 in the revised manuscript)

Agley CC, Velloso CP, Lazarus NR, Harridge SD. An image analysis method for the precise selection and quantitation of fluorescently labeled cellular constituents: application to the measurement of human muscle cells in culture. J Histochem Cytochem. 2012 Jun;60(6):428-38. doi: 10.1369/0022155412442897.

  1. There are also other problems with figure 3. The regenerating zones are not outlined, and it is not clear whether identical areas were used to compare the numbers of myofibers with centralized nuclei. In addition, images in figure 3b, e, and h were not properly stained, and it is not possible to see myofibers.

Responses: Thank you for the comments. Figure 3 (Figure 2 after revision) has been replaced by images in a higher magnification to show the centralized nuclei more obviously. The whole area of each image was the damaged/regenerating zone so that the regenerating zones could not be outlined further in the images. Scale bars were integrated in each image so that the areas to be compared were identical. Images in figure 3b, e, and h have been replaced.

  1. The rational to compare CDR and VTR in muscle regeneration is not clear.

Response: Thank you for the comments. VTR was used to act as the positive control in the in vivo experiment. The explanation has been added in the Materials and Methods (Section 2.3).

  1. In figures 4 and 5, it is not clear how many regenerating myofibers were examined in each condition.

Response: Thanks for your comment. Immunostaining images have been added (Figure 3 to Figure 5 in the revised manuscript) so that the number of myofibers could be shown.

  1. The authors should show immunostaining of MyoD and MyoG in regenerating myofibers.

Response: Thanks for the suggestion. Immunostaining images of MyoD and MyoG have been added (Figure 4 and Figure 5 in the revised manuscript).

  1. There are some inconsistent results in RT-qPCR analyses of gene expression. Figure 5A shows that CDR induces more MyoD-positive nuclei than VTR at 2 days post-injury. However, figure 6B shows that CDR is less efficient to induce MyoD expression than VTR. Similarly, figure 5B shows more MyoG-positive nuclei at 2 days after CRD and VTR treatment, but figure 6B shows reduced expression of MyoG in both conditions.

Response: Thank you for the comment. These non-coherent results might be caused by the differences in the region of investigation in the two assessments. images captured specifically at the central damaged zone were used for the IHC analysis, while the whole contused region of the gastrocnemius muscle (including the damaged zone, border zone and the non-damaged tissue) was harvested for the qPCR analysis. This explanation, together with a reference support, have been added in the Discussion.

  1. Figure 2 should be moved to supplementary materials, while figure 4S should be shown in the main text, but the quality and magnification of the images should be improved.

Response: Thank you for your suggestion. Figure 2 has moved to supplementary materials and the Figure 4S is moved to the main text (Figure 3A in the revised manuscript) after its quality and magnification have been improved.

Comments on the Quality of English Language

The rational for some experiments should be better explained, and some descriptions in the method section need to be revised.

Responses: Thanks for the comments. The manuscript has been revised extensively.

Round 2

Reviewer 1 Report

The manuscript by Sui et al. has been improved by revision. However, I still find some minor points to be addressed:

Abstract:

Line 14: Please replace ”, abbreviated as CDR,” simply by (CDR).

Lines 14-15: In the current version of the Abstract, the animal model is introduced first, but the results from the animal model follow in vitro experiments. This is confusing. The in vitro cytotoxicity experiments should be explained before the animal model and its results.

Line 14: Please replace muscle cells by immortalized rat myoblast culture.

Line 96: Please revise “herbal pastes in a dry weight ratio of 1:1:1.”.

Line 151: Please remove “(abbreviated from its pronunciation)”. It is sufficient to mention Voltaren and use (VTR) further in the text. However, the authors may add the trademark symbol ™ by the first stating.

Line 157: Please remove the space.

Lines 168-174: Please remove italic style.

Line 174: Please use brackets when referring to a section number as (2.5.1.).

Line 247: Please remove (0 µg/ml).

Line 248: Please refer to x axis …different concentrations (x axis).

Line 250:  Please revise n=6 (3.2.).

Figure 4J: Please relocate the ### on the top of the respective lines.

Line 473: Please revise the number of Supplementary Figure 4S to S1. There are no other Supplementary Figures. Besides, it was referred to this figure in line 252 as Figure S1.

The manuscript text should undergo further revision and proofreading by our English Editing department before publication.

Author Response

Comments and Suggestions for Authors

The manuscript by Sui et al. has been improved by revision. However, I still find some minor points to be addressed:

Abstract:

Line 14: Please replace ”, abbreviated as CDR,” simply by (CDR).

Response: Revised as suggested.

Lines 14-15: In the current version of the Abstract, the animal model is introduced first, but the results from the animal model follow in vitro experiments. This is confusing. The in vitro cytotoxicity experiments should be explained before the animal model and its results.

Response: Thank you for your suggestion. The in vitro cytotoxicity experiment has been explained before the animal model and its results in the Abstract (Line12-13).

Line 14: Please replace muscle cells by immortalized rat myoblast culture.

Response: Thank you for your suggestion. They were replaced as suggested.

Line 96: Please revise “herbal pastes in a dry weight ratio of 1:1:1.”.

Response: Revised as your suggestion.

Line 151: Please remove “(abbreviated from its pronunciation)”. It is sufficient to mention Voltaren and use (VTR) further in the text. However, the authors may add the trademark symbol ™ by the first stating.

Response: Thanks for the suggestion. The sentence has been revised accordingly.

Line 157: Please remove the space.

Response: Removed as your request.

Lines 168-174: Please remove italic style.

Response: Sorry for the careless mistake. This paragraph has been reformatted.

Line 174: Please use brackets when referring to a section number as (2.5.1.).

Response: “2.5.1” is the subheading number of “Immunofluorescence”. It has been corrected. Sorry for the careless mistake.

Line 247: Please remove (0 µg/ml).

Response: Removed as your request.

Line 248: Please refer to x axis …different concentrations (x axis).

Response: “(x axis)” has been added.

Line 250:  Please revise n=6 (3.2.).

Response: Sorry for the careless mistake. It has been revised.

Figure 4J: Please relocate the ### on the top of the respective lines.

Response: Thank you for the suggestion. Figure 4J has been revised.

Line 473: Please revise the number of Supplementary Figure 4S to S1. There are no other Supplementary Figures. Besides, it was referred to this figure in line 252 as Figure S1.

Response: Sorry for the careless mistake. It has been revised.

Comments on the Quality of English Language

The manuscript text should undergo further revision and proofreading by our English Editing department before publication.

Response: The manuscript text has been undergone further vision and proofreading.

Reviewer 2 Report

The authors have addressed most issues raised in my review.

Author Response

The authors have addressed most issues raised in my review.

Response: Thank you very much for your comments and suggestions for us to improve our manuscript.